# Mapping factors that may influence attrition and retention of midwives: a scoping review protocol

Gill Moncrieff ,[1] Soo Downe ,[2] Margaret Maxwell ,[1] Helen Cheyne [1]

¹University of Stirling, Stirling, UK
²University of Central Lancashire, Preston, UK

**Correspondence to**
Gill Moncrieff;
Gill.Moncrieff@stir.ac.uk

## ABSTRACT

**Introduction** An appropriately staffed midwifery workforce is essential for the provision of safe and high-quality maternity care. However, there is a global and national shortage of midwives. Understaffed maternity services are frequently identified as contributing to unsafe care provision and adverse outcomes for mothers and babies. While there is a need to recruit midwives through pre-registration midwifery programmes, this has significant resource implications, and is counteracted to a large extent by the high number of midwives leaving the workforce. It is increasingly recognised that there is a critical need to attend to retention in midwifery in order to develop and maintain safe staffing levels. The objective of this review is to collate and map factors that have been found to influence attrition and retention in midwifery.

**Methods and analysis** Joanna Briggs Institute guidance for scoping reviews and the Preferred Reporting Items for Systematic Reviews and Meta-Analyses extension for Scoping Reviews will be used to guide the review process and reporting of the review. CINAHL, MEDLINE, PsycINFO and Scopus databases will be searched for relevant literature from date of inception to 21 July 2023. Research from high-income countries that explores factors that influence leaving intentions for midwives will be included. Literature from low-income and middle-income countries, and studies where nursing and midwifery data cannot be disaggregated will be excluded. Two reviewers will screen 20% of retrieved citations in duplicate, the first author will screen the remaining results. Data will be extracted using a preformed data extraction tool by the first author. Findings will be presented in narrative, tabular and graphical formats.

**Ethics and dissemination** The review will collate data from existing research, therefore ethics approval is not required. Findings will be published in journals, presented at conferences and will be translated into infographics and other formats for online dissemination.

## STRENGTHS AND LIMITATIONS OF THIS STUDY

⇒ Scoping reviews provide a rigorous and structured method through which to collate and map evidence on a given topic.
⇒ This protocol and the full review will follow Joanna Briggs Institute guidance for scoping reviews and will be reported in line with the Preferred Reporting Items for Systematic Reviews and Meta-Analyses extension for Scoping Reviews.
⇒ Relevant research that has not been listed in the included databases may not be identified for inclusion in the review.
⇒ It may be difficult to establish whether participants intentions to leave or stay relate to changing role, changing organisation or leaving the profession altogether.
⇒ The review will be of relevance to other high-income countries but is unlikely be relevant for low-income and middle-income countries.

## INTRODUCTION

The provision of safe, effective and quality maternity services is essential for the health and well-being of women and babies.[1] Midwives are health professionals that are appropriately educated and registered according to the standards of the International Confederation of Midwives,[2] and are integral to sexual, reproductive, maternal, newborn and adolescent health.[3] A midwifery workforce that is appropriately and sustainably staffed is integral to this. However, there is a national and global shortage of healthcare professionals, with nurses and midwives at the top of the list among the healthcare professions, representing over 50% of the global shortage.[4] For maternity care, staff shortages appear increasingly to impact on safety.

Several recent maternity investigations and reviews from the UK have identified understaffing as a contributory cause in adverse outcomes for mothers and babies.[5–8] The Care Quality Commission frequently finds insufficient staffing in maternity units, which they report is putting mothers and babies at risk.[9–11] Both the recruitment and retention of staff are contributing to this problem. A primary action outlined in the recent Ockenden review of maternity services at Shrewsbury and Telford Hospital NHS Trust is to ensure sufficient staffing for the provision of safe and sustainable maternity systems.[8]

In 2021, the Health and Social Care Committee (HSCC) reported that the

National Health Service (NHS) in England is short of 1932 midwives, describing staffing shortages in maternity as a persistent problem.[12] While there is a need to recruit new midwives to ameliorate this, training, recruitment and selection processes come with considerable time and cost implications. Since the HSCC report, the number of midwives in England has fallen by a further 633 full time equivalent posts between April 2021 and April 2022. This is reportedly the largest annual loss of midwifery staff from the NHS since 2009, when records for this measure were first recorded.[9]

Historically, the UK has relied on bringing new health professionals into the workforce to deal with staff shortages, whether through educating new health professionals, or looking to international recruitment.[13 14] However, there are ethical issues associated with international recruitment,[15] and education and training packages are required due to differing training practices between countries. Moreover, staff from minority ethnic groups have received poor treatment in the past.[14 16] This, along with the UK's exit from the European Union may encourage foreign-trained professionals to choose other countries rather than the UK. Undergraduate training for health professionals in the UK is associated with significant costs and places on courses are finite, particularly given staffing shortages both in practice and educational establishments.[17] Furthermore, applications for places on Nursing and Midwifery courses have fallen over recent years and attrition from nursing and midwifery degrees is significant (at 24% and 21% of the student intake, respectively).[18]

Equally important, loss of staff from the existing workforce results in the loss of valuable experience. It also has cost and resource implications and reduces productivity and quality of care of care provision.[19] The NHS in general is experiencing ongoing and increasing difficulty, in many areas, with retaining its existing staff, a process, that is, critical to the effective functioning and sustainability of any organisation.[18 20] Retention also represents a faster and less costly way to maintain the workforce than relying on new recruits. The need for a focus on retention and the development of strategies to increase retention for healthcare workers is increasingly recognised as critical both to attend to the current staffing crises and to facilitate long-term stability and productivity of the healthcare workforce.[12 18 20] This may be increasingly necessary following the experiences of staff over the COVID-19 pandemic, which may have tipped the balance further towards an exodus of staff from the service.[20 21]

The objective of the review is to collate and map the factors that have been found to influence attrition and retention in midwifery. It forms part of a larger project funded by the Scottish Government Chief Scientist Office, that is, designed to develop a strategy to increase retention within the UK midwifery workforce (the REMAIN study). Following completion of the review, the findings will be used to collaborate with the REMAIN stakeholder groups to identify key questions for subsequent stages of the research and to feed into development of the retention strategy.

The social ecological model developed by McLeroy *et al*[22] will be used as a framework for analysis and presentation of findings. An ecological approach to the analysis is systems-oriented, facilitating consideration of the role of the causal processes operating in and across the different system levels and the relationships within and between these.[23] This moves the focus away from individual causes of problems towards multifactor environmental causes, and thus multilevel systems-focused solutions.

It is recognised that intention to leave and to intention to stay are not mirror constructs, and that influences on intention may differ from factors that influence the act of leaving.[19 24] Furthermore, the decision to stay or leave may include changing role, changing organisation or leaving the profession altogether.[24] Therefore, analysis and the resulting framework will separate out these constructs where this is possible.

A scoping review was considered an appropriate method for this review which does not aim to synthesise the findings, rather the objective is to collate and map the factors identified as influencing attrition and retention and present these findings in a clearly illustrated tabular and/or graphical format. Scoping review methodology provides a rigorous and structured approach through which to achieve this.[25 26]

### Review questions

The main research question is:
► What factors influence midwives' intention or decision to stay or leave?

Secondary research questions are:
► What associated recommendations have been made to improve retention in midwifery?
► What gaps need to be filled to make recommendations for research, policy and practice?

### Review registration

This review protocol has been registered with Open Science Framework.

### METHODS AND ANALYSIS

The review will be carried out according to the Joanna Briggs Institute (JBI) guidance for scoping reviews[26] and will be structured according to the Preferred Reporting Items for Systematic Reviews and Meta-Analyses (PRISMA) extension for Scoping Reviews,[25] both of which have guided the reporting of this protocol.

A preliminary search of PROSPERO and CINAHL search (18 May 2023) confirmed that there are currently no existing or in progress systematic or scoping reviews that collate the factors that influence midwives' motivation to stay in or leave their role.

**Table 1** Search terms

| Search ID | Search terms | Results |
|---|---|---|
| S1 | (MH "Midwives+") OR (MH "Midwife Attitudes") OR (MH "Midwifery+") OR (MH "Nurse Midwifery") OR (MH "Midwifery Service+") OR (MH "Nurse-Midwifery Service") OR (MH "Maternal Health Services+") | 53 880 |
| S2 | TI (midwi* or (maternity N3 service*) OR AB (midwi* or (maternity N3 service*) | 41 052 |
| S3 | (MH "Personnel Retention") OR (MH "Personnel Loyalty") OR (MH "Employment Termination") | 15 649 |
| S4 | TI (work* or profession* or employ* or occupation* or role* or organisation* or position or career* or vacanc*) N5 (retention or retain* or remain* or stay* or leav* or quit* or resign* or attrition or turnover) OR AB (work* or profession* or employ* or or occupation* or role* or organisation* or position or career* or vacanc*) N5 (retention or retain* or remain* or stay* or leav* or quit* or resign* or attrition or turnover) | 23 722 |
| S5 | S3 OR S4 | 37 028 |
| S6 | S1 OR S2 | 57 065 |
| S7 | S5 AND S6 | 429 |

## Information sources and searches

Databases will be searched from date of inception to 21 July 2023. An initial limited search of MEDLINE was carried out to identify relevant articles to develop the full search strategy. Search terms for the full search strategy were identified based on the titles, abstracts and index terms used to describe the articles. The search strategy for the initial database was then developed and tested with an information specialist. Table 1 outlines the search strategy developed for CINAHL (via EBSCO). This will be adapted for each of the databases to be used in the full review. A full search will then be carried out using MEDLINE, MIDRS and Scopus databases (online supplemental file 1). A second search will be carried out through screening the reference lists of all papers included in the review. Finally, the websites of professional bodies will be searched, and relevant professionals will be contacted to identify grey literature for inclusion. As stated in JBI guidance, it is possible that additional keywords, search terms or information sources may be identified as the search commences.[26] If this is the case, amendments to the search strategy will be made transparent in the full review.

## Eligibility criteria

JBI guidance defines eligibility according to participants, concept and context.[26]

## Participants

Midwives as defined by the International Confederation of Midwives.[2] This includes midwives that have practiced or practice within a healthcare, education, research, or policy setting and privately practicing and independent midwives. Where publications include both nurses and midwives, and data for midwives can be disaggregated, these will be included. However, if responses from midwives cannot be separated, these publications will not be included.

## Concept

Factors that influence midwives' intention or decision to stay in or leave their role as a midwife. This will include factors that influence whether midwives move from one role or organisation to another, as well as factors that influence the intention or decision to stay in or leave the profession entirely. Only research where the primary focus is the decision to stay or leave will be included. Research that has another focus, but that may have a decision to stay or leave as an outcome (eg, research focused on well-being), will be excluded. Where studies measure leaving intention quantitatively, but do not explore the associated reasons, these will also be excluded from the analysis.

## Context

High-income countries as classified by the World Bank.[27] Studies from low-income and middle-income countries will be excluded, in order to focus on perspectives and experiences that derive from a similar context to that of the UK maternity system. It is recognised that even with this restriction, contexts that are considered to be disparate to the UK may be eligible for inclusion. Where this is the case, any distinctions will be included in the analysis and documented in the findings.

## Types of studies

All primary (qualitative, quantitative and mixed methods) research studies will be eligible for inclusion. Reviews will not be eligible for inclusion to avoid duplication of the included studies, but their reference lists will be screened for relevant primary research papers. Relevant grey literature will also be included, for example, surveys carried out by professional bodies that may not have been published in journals. Conference abstracts and other non-full text publications will not be eligible for inclusion. There will be no language or date restrictions on the search. Google translate will be used for translation of any non-English language publications.

## Study screening and selection

Following the database search, the retrieved citations will be uploaded to Rayyan and duplicates removed. Citations will be screened initially by title and abstract, then by full text using the inclusion/exclusion criteria (table 2). Where articles are excluded at the full text stage, the reason will be recorded on Rayyan and documented in the full review. To ensure consistency within the review

**Table 2** Eligibility criteria

| | Inclusion | Exclusion |
|---|---|---|
| Participant | Midwives that practice or have practiced in:<br>► Healthcare (including privately practicing/ independent midwives).<br>► Education.<br>► Policy.<br>► Research. | Publications that include nurses and midwives, where data for midwives cannot be disaggregated. |
| Concept | Decision or intention to:<br>► Stay in or change role.<br>► Stay in or change organisation.<br>► Stay in or leave the profession. | Publications where decision to stay in or leave midwifery is not the primary focus of the research. |
| Context | High-income countries. | Middle-income countries.<br>Low-income countries. |

team regarding inclusion/exclusion criteria, 20% of the retrieved citations will be screened by two reviewers at both the title/abstract and full text stage. Any disagreements will be discussed within the team. The first author will screen the remaining results once consensus has been reached. The screening and selection process will be reported in a PRISMA-Scoping Reviews flow diagram.

### Data extraction

Once the articles for inclusion have been selected, data will be extracted onto an Excel document by the first author, using a data extraction tool developed for the purposes of this review (online supplemental file 2). This tool may be modified as necessary as data are extracted. If this does occur, amendments will be detailed in the full review. The following details will be extracted: authors; publication date; title; location of study; aims/objectives; number of participants; study design; area of midwifery practice; factors found to influence the decision to stay or leave; and recommendations for policy, research or practice. The data extraction tool has been developed in collaboration with the review team and the first author will discuss any queries, concerns or potential modifications during the data extraction process with the rest of the team.

Critical appraisal of included studies is not required or usually included as part of the review process for scoping reviews.[26] This is due to the stated purpose of describing and mapping the evidence, rather than making analytical comparisons and/or producing evidence to directly inform practice.

### Data analysis and presentation

Extracted data will be reviewed and discussed by the review team. Data will be summarised narratively and in a tabular and graphical format. These will focus on the main objective of the review, to summarise and illustrate

factors found to influence attrition and retention. Recommendations for research and practice and gaps in the research will also be documented.

### Patient and public involvement

The REMAIN project includes engagement with staff, service users and expert members of its Intervention Development Group to inform research processes and refine findings. Through such stakeholder engagement, the findings of the review will be prioritised to feed into the development of a retention strategy for midwives.

### Ethics and dissemination

The review will collate data from existing research, therefore ethics approval is not required. Findings will be published in journals, presented at conferences and will be translated into infographics and other formats for online dissemination.

## DISCUSSION

Sustainable staffing levels are integral to the provision of safe and quality maternity care. This requires appropriate retention of midwives within the workforce. This will be the first review to systematically map the factors that have been found to influence midwives' leaving intentions, providing valuable information to feed into retention-related activities and guidelines for policy and practice. Ultimately it will inform the development of an evidence-based retention programme that aims to reduce attrition within the midwifery workforce.

A scoping review is the appropriate method for this review where the objective is to collate and map the factors that have been found to influence attrition and retention in midwifery, rather than synthesising findings or making analytical comparisons.[26] Stakeholder engagement with the REMAIN Intervention Development Group will prioritise findings to identify recommendations for practice and to feed into the development of a retention strategy for midwives. This engagement exercise will also identify gaps to be explored as part of the research.

JBI guidance for scoping reviews has been followed to develop this protocol and will be used for the full review, providing a rigorous and structured method to collate and map the evidence relating to midwives leaving intentions.[26] It is however recognised that there may be some research, such as workforce surveys carried out by relevant professional bodies, that is, not listed in bibliographic databases. To counter this, we will carry out additional searches, including screening reference lists of included publications, searching the websites of professional bodies and contacting relevant professionals. However, it is possible that relevant research may not be picked up through these additional searches.

In addition, retention is a concept that has yet to be consistently defined and understood, both within the relevant literature and in research, policy and practice.[19 24] As a result, it is likely that it will be challenging to

determine whether the included data refers to midwives' intentions to move to another role in the same organisation, or to move to a different organisation, or whether they intend to leave the profession altogether. Where possible however, data for these particular constructs will be separated and clarified in the analysis and findings of the review.

Only data from high-income countries will be eligible for inclusion, with the intention of producing findings that are relevant to the UK and other similar maternity systems. However, it is recognised that this distinction is not necessarily clear and that we may as a result include data that relate to divergent maternity systems. The context for each included study will however be recorded and if this issue arises, it will be discussed within the review. A separate review will be required for low-income and middle-income settings.

This review is timely and highly necessary to inform retention-related activities. Working conditions in maternity have been found to have a profound and detrimental impact on the mental health and leaving intentions of midwives, and on the quality and safety of care provision.[28 29] As midwives leave, this exacerbates these issues, resulting in a vicious cycle of staff attrition. As a result, there has been considerable loss of valuable experience from the workforce, compromising the safety of care provision and the clinical education our future midwives.[12 30] This situation cannot be resolved simply by adding more midwives. There is a much-needed commitment to attend to the underlying factors that motivate leaving intentions, many of which are modifiable with resolve.[29] This review will identify and collate these factors, providing valuable evidence with which to begin this endeavour.

**Acknowledgements** With thanks to Joshua Cheyne and Catherine Harris for support with formulating the search terms for the review.

**Contributors** GM conceived and designed the study and first draft of this protocol. HC, SD and MM critically reviewed and advised on the first draft. All authors contributed to the final draft of the protocol.

**Funding** This work was supported by the Chief Scientist Office, which is part of the Scottish Government Health Directorates. Grant number: CAF/23/06.

**Competing interests** None declared.

**Patient and public involvement** Patients and/or the public were involved in the design, or conduct, or reporting, or dissemination plans of this research. Refer to the Methods section for further details.

**Patient consent for publication** Not applicable.

**Provenance and peer review** Not commissioned; externally peer reviewed.

**ORCID iDs**
Gill Moncrieff http://orcid.org/0000-0001-7142-9953
Soo Downe http://orcid.org/0000-0003-2848-2550
Margaret Maxwell http://orcid.org/0000-0003-3318-9500
Helen Cheyne http://orcid.org/0000-0001-5738-8390

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
