## [Reviewer comments · BMJ Open]

ARTICLE DETAILS

TITLE (PROVISIONAL)	Mapping factors that may influence attrition and retention of midwives: a scoping review protocol
AUTHORS	Moncrieff, Gill; Downe, Soo; Maxwell, Margaret; Cheyne, Helen

VERSION 1 – REVIEW

REVIEWER	O'Connell, Maeve Fatima College of Health Sciences, Midwifery
REVIEW RETURNED	23-Jul-2023

GENERAL COMMENTS	BMC Pregnancy Title: Mapping factors that may influence attrition and retention of midwives: a scoping review protocol Many thanks for the opportunity to review this review protocol which is well written and justified overall. The findings of the full review will be useful and important. Therefore it is important that the review protocol is robust. I find the paper should be suitable for publication pending a few minor changes. The introduction is well written and supported by relevant evidence providing a justification for the review and is an interesting topic. Since you have in the exclusion criteria about studies which include nurses and midwives where data for midwives can not be disaggregated, in the very beginning of the introduction I think you need to define what is a midwife to make it really clear who is eligible – ICM definition of a midwife? Page 4 Line 3. “The objective of the review...” This is rather the aim as it is a broad statement. Objectives should be specific and measurable if using. I suggest it might be worthwhile to include objectives as well as the general aim. I agree that the ecological model is a relevant framework for the analysis. The search strategy looks well developed and makes sense. Minor: Review questions, include a question mark for each question. Page 6 Line 3. Spelling error: countries Page 6. Line 6. ‘felt’ suggest change to ‘considered’. I never heard of Rayyan and looked it up – it looks great! Good luck using it!
--

	Eligibility criteria is clear. Suggestion for data extraction – area of midwifery practice- it would be worth noting if the midwife works in clinical/ education/ research or policy. Best wishes with this great work.
--	--

REVIEWER	HakemZadeh, Farimah York University
REVIEW RETURNED	31-Jul-2023

GENERAL COMMENTS	Thank you for preparing the Scoping Review Protocol for mapping factors that may influence the attrition and retention of midwives. As mentioned in the abstract and introduction of the review protocol, the global and national shortage of midwives results in inadequate staffing and the inability of healthcare systems to adequately address the care of birthing parents and newborns. Therefore mapping the factors that can influence the attrition and retention of midwives can help researchers and policymakers in developing interventions for the retention of midwives in the health workforce. I hope the points provided here assist you in improving the review protocol and conducting a thorough scoping review.  1. The search strategy eliminates closely relevant constructs of organizational commitment and career commitment. The review protocol can benefit from an explanation of why these relevant constructs are not included among the search term to answer the main question of the review. This discussion can expand on the distinction between intention to stay/intention to leave and career/organizational commitment. The Theory of Planned Behaviour might be beneficial in developing this discussion. This will be beneficial in later making informed and structured inclusion and exclusion decisions as well. 2. The protocol similarly eliminated studies that have directly measured retention rather than the intention to leave or stay. Please explain whether those studies will be included or excluded and why. 3. For the study screening and selection, the size of the review team and the division of labour should be clarified. Good luck, and looking forward to reading the full scoping review.
---

VERSION 1 – AUTHOR RESPONSE

Reviewer 1 revisions	
Comments	Responses
Since you have in the exclusion criteria about studies which include nurses and midwives where data for midwives cannot be disaggregated, in the very beginning of the introduction I think you need to define what is a midwife to make it really clear who is eligible –	We have now included the ICM definition of a midwife at the beginning of the protocol.

ICM definition of a midwife?	
Page 4 Line 3. "The objective of the review..." This is rather the aim as it is a broad statement. Objectives should be specific and measurable if using. I suggest it might be worthwhile to include objectives as well as the general aim.	Thank you for making this point. We appreciate that the overall aim is usually stated, but objectives is the term that is stipulated in JBI guidance to describe the purpose of the review (Peters 2022 in the reference list).
Review questions, include a question mark for each question	Thank you, we have now done this.
Page 6 Line 3. Spelling error: countries	Thank you, this has now been corrected.
Page 6. Line 6. 'felt' suggest change to 'considered'	Thank you, this has now been changed as suggested.
Suggestion for data extraction – area of midwifery practice- it would be worth noting if the midwife works in clinical/ education/ research or policy.	We have now included this as one of the details that will be extracted from included literature.

Reviewer 1 revisions	
Comments	Responses
The search strategy eliminates closely relevant constructs of organizational commitment and career commitment. The review protocol can benefit from an explanation of why these relevant constructs are not included among the search term to answer the main question of the review. This discussion can expand on the distinction between intention to stay/intention to leave and career/organizational commitment. The Theory of Planned Behaviour might be beneficial in developing this discussion. This will be beneficial in later making informed and structured inclusion and exclusion decisions as well.	Thank you for this suggestion. The review protocol is designed to capture all studies focused on why midwives leave or stay, with no prior pre-conceptions of what might be found to be the underpinning theories or mechanisms that have already been explored. It is possible that some of these studies explore concepts such as organisational commitment or career commitment (or other issues such as the impact of organisational ethos on decision making) but we do not want to pre-determine these at this stage. We agree that the theory of planned behaviour is a powerful theoretical construct, and we have used it in previous studies and reviews. However, again, in this case we are undertaking this scoping review to see what is out there so we have made the decision not to frame the review with an explicit a priori theoretical stance at this point.
The protocol similarly eliminated studies that have directly measured retention rather than	Thank you for this comment. We have now clarified that studies that measure leaving

the intention to leave or stay. Please explain whether those studies will be included or excluded and why.	intention quantitatively, but do not explore the associated reasons, will be excluded. The reason for exclusion is that we want to collate the reasons rather than the numbers.
For the study screening and selection, the size of the review team and the division of labour should be clarified.	We have clarified how many members of the review team will be involved in screening and selection.

VERSION 2 – REVIEW

REVIEWER	O'Connell, Maeve Fatima College of Health Sciences, Midwifery
REVIEW RETURNED	04-Sep-2023

GENERAL COMMENTS	Well done on considering the feedback. I recommend the manuscript for publication and wish the team all the best with the review, I look forward to reading it.
---

REVIEWER	HakemZadeh, Farimah York University
REVIEW RETURNED	05-Sep-2023

GENERAL COMMENTS	Thank you for addressing the comments on the revised manuscript.
--